# Radiomics-Based Predictive Model of Radiation-Induced Liver Disease in Hepatocellular Carcinoma Patients Receiving Stereo-Tactic Body Radiotherapy

**DOI:** 10.3390/biomedicines10030597

**Published:** 2022-03-03

**Authors:** Po-Chien Shen, Wen-Yen Huang, Yang-Hong Dai, Cheng-Hsiang Lo, Jen-Fu Yang, Yu-Fu Su, Ying-Fu Wang, Chia-Feng Lu, Chun-Shu Lin

**Affiliations:** 1National Defense Medical Center, Department of Radiation Oncology, Tri-Service General Hospital, Taipei 114, Taiwan; 31235@mail.ndmctsgh.edu.tw (P.-C.S.); hwyyi@mail.ndmctsgh.edu.tw (W.-Y.H.); 496010464@mail.ndmctsgh.edu.tw (Y.-H.D.); lsir183@mail.ndmctsgh.edu.tw (C.-H.L.); ugidgal@mail.ndmctsgh.edu.tw (J.-F.Y.); m871462@mail.ndmctsgh.edu.tw (Y.-F.S.); doc31006@mail.ndmctsgh.edu.tw (Y.-F.W.); 2Department of Biomedical Imaging and Radiological Sciences, National Yang Ming Chiao Tung University, Taipei 112, Taiwan; 3Institute of Clinical Medicine, National Yang-Ming Chiao Tung University, Taipei 114, Taiwan; 4National Defense Medical Center, Institute of Medical Science, Taipei 114, Taiwan

**Keywords:** radiation-induced liver disease, stereotactic body radiation therapy, radiomics, predictive model, decision making

## Abstract

(1) Background: The application of stereotactic body radiation therapy (SBRT) in hepatocellular carcinoma (HCC) limited the risk of the radiation-induced liver disease (RILD) and we aimed to predict the occurrence of RILD more accurately. (2) Methods: 86 HCC patients were enrolled. We identified key predictive factors from clinical, radiomic, and dose-volumetric parameters using a multivariate analysis, sequential forward selection (SFS), and a K-nearest neighbor (KNN) algorithm. We developed a predictive model for RILD based on these factors, using the random forest or logistic regression algorithms. (3) Results: Five key predictive factors in the training set were identified, including the albumin–bilirubin grade, difference average, strength, V5, and V30. After model training, the F1 score, sensitivity, specificity, and accuracy of the final random forest model were 0.857, 100, 93.3, and 94.4% in the test set, respectively. Meanwhile, the logistic regression model yielded an F1 score, sensitivity, specificity, and accuracy of 0.8, 66.7, 100, and 94.4% in the test set, respectively. (4) Conclusions: Based on clinical, radiomic, and dose-volumetric factors, our models achieved satisfactory performance on the prediction of the occurrence of SBRT-related RILD in HCC patients. Before undergoing SBRT, the proposed models may detect patients at high risk of RILD, allowing to assist in treatment strategies accordingly.

## 1. Introduction

According to the 2017 Cancer Registry Annual Report of the Health Promotion Administration of Taiwan, primary liver carcinoma (PLC) accounts for 10% of newly diagnosed cancers and 17.5% of cancer-related deaths. In addition, PLC is the second leading cause of cancer-related deaths worldwide [1], and hepatocellular carcinoma (HCC) accounts for approximately 85% of PLC cases.

Fewer than 30% of patients are eligible for curative surgery or transplantation at the time of HCC diagnosis [2,3]. In such inoperable cases, radiofrequency ablation (RFA), transarterial chemoembolization (TACE), and radiotherapy (RT) have been used as alternative options. In recent decades, with advances in radiotherapy techniques, the use of stereotactic body radiotherapy (SBRT) to deliver high radiation doses to the tumor has become a feasible and attractive option. However, the tolerance of a normal liver and related complications is still a constriction in the use of SBRT. Radiation-induced liver disease (RILD) is a common complication of SBRT without an effective treatment [4]. Most previous studies have identified baseline liver function as an important risk factor of RILD; however, there is no strong consensus nor consistent results on the role of conventional RT nor SBRT features, such as dose-volumetric parameters for RILD prediction [5,6,7,8,9,10]. Some studies reported that some dose-volumetric parameters, including liver mean dose, effective liver volume, and doses to 700–900 cc, were associated with liver toxicity, but other studies did not.

Radiomics is a method that mines features from radiographic medical images; these features may be subsequently analyzed to quantitatively characterize a disease. Radiomics has attracted significant research attention in the field of clinical medicine due to its potential for screening, diagnosis, and prognosis [11,12,13], including the staging of liver fibrosis [14]. These findings indicate that radiomic features based on the computed tomography (CT) image obtained before RT may provide information about baseline liver function and may be associated with the risk of RILD.

To date, models for accurately estimating the risk of RILD in patients with hepatocellular carcinoma are limited. Thus, the present study aimed to develop a model for predicting the occurrence of RILD, based on a combination of clinical, dose-volumetric, and radiomic features.

## 2. Materials and Methods

### 2.1. Patients

Medical records of HCC patients treated with SBRT in our institution between July 2007 and June 2015 were retrospectively reviewed. The inclusion criteria were as follows: (1) Dynamic CT images obtained before SBRT available for the analysis of radiomic features, (2) dose-volumetric parameters were available, and (3) follow-up data available and follow-up time for non-RILD patients longer than 4 months. The final dataset for the subsequent analyses included 86 patients.

These patients had a confirmed diagnosis, and their treatment options were considered by a multidisciplinary team composed of experienced oncologists and radiologists. The most common treatment indications were HCC, which was medically inoperable or unresectable and unsuitable for TACE or RFA, and patients refused to undergo other locoregional therapies.

This study was approved by our institutional review board; the informed consent requirement was waived due to the retrospective nature of the present study.

### 2.2. SBRT

SBRT preparation, technique, and dose constraints are described in our previous publication [15]. In brief, all patients were treated using the CyberKnife^®^ system (Accuracy Inc., Sunnyvale, CA, USA) with real-time tumor tracking devices. Most patients received fiducial marker implantation for the tumor tracking technique. The gross tumor volume (GTV) was defined as a radiographically visible tumor based on contrast-enhanced CT or magnetic resonance imaging (MRI) scans; the clinical target volume (CTV) was equivalent to the GTV. The planning target volume (PTV) was obtained by adding 0–8 mm of a margin to the corresponding CTV; this was modified when dose-limiting organs overlapped, except for the normal liver. We tended to give the highest dose to the gross tumor under the allowable dose constraints. For patients with larger tumor size (>6 cm), Child–Pugh score 7–8, or multiple tumors (≥3), we might give a more conservative dose. Treatment was administered in 2–6 fractions, with a total dose of 25–60 Gy (median 45 Gy) prescribed to the PTV. The median, mean, and range of estimated equivalent dose in 2 Gy fractions (EQD2) with α/β = 10 Gy were 71.2, 70.2, and 36.4–110.0 Gy, respectively. However, caution should be taken that linear-quadratic model may overestimate the EQD2 when dealing with a large fraction size (>5 Gy per fraction).

### 2.3. Patient Follow-Up and Definition of RILD

All patients underwent follow-up consultation, liver function tests, and abdominal dynamic CT and/or MRI scans at 1–4 months after the completion of SBRT, and at 3-month intervals thereafter for outcome and side effect evaluation. Patients who experienced disease progression received salvage or palliative therapy.

Classic RILD was defined as elevated levels of alkaline phosphatase more than twice over the upper normal limit within 4 months [5]. Non-classic RILD was defined as elevated levels of liver transaminases more than five times over the upper limit within 4 months [4] or worsening of the Child–Pugh (CP) score by 2 or more points within 3 months [9]. Patients with progressive HCC were not diagnosed with RILD even if they met these criteria. In the present study, we included both the classic and non-classic RILD patients according to above criteria.

### 2.4. Extraction of Radiomic Features

The radiomics feature extractions were performed only for the region of interest (ROI) on the simulation CT. None of the follow-up images were used to calculate the radiomics features. Our region of interest (ROI) in this study was the normal liver in the arterial phase. It included the total liver volume minus the GTV, any cysts, and the gall bladder. The ROIs were initially contoured by the medical physicist on the simulation CT, and it would be modified by the doctor based on the consensus of two radiation oncologists. We then saved these images in the format of Digital Imagine and Communications in Medicine. We used the 3D Slicer software which is based on the py-radiomics module to extract radiomic features [16]. Image resolution was resampled to the isotropic voxel size of 1 × 1 ×1 mm^3^ by applying the interpolation method for further calculation of 3D radiomic features. Because there was an absolute Hounsfield unit of CT image, we fixed the bin size at 25 to maintain the direct relationship with the original intensity scale. So, the intensity within the delineated normal liver volume would be discretized into 5 bins, which was suitable for the relatively homogenous organ under contrast CT including liver.

Overall, 107 radiomic features including 16 first-order statistics, 16 shape-based, 24 gray-level co-occurrence matrixes (GLCM), 16 gray-level run-length matrixes (GLRLM), 16 gray-level size-zone matrixes (GLSZM), 14 gray-level dependence matrixes (GLDM), and 5 neighboring gray-tone difference matrixes (NGTDM), were extracted from each patient. All the image preprocessing steps and subsequent radiomics extractions were performed and complied with the Image Biomarker Standardization Initiative (IBSI) reference manual [17].

### 2.5. Selection of Predictive Factors

We selected predictive factors from three aspects, which contained clinical, radiomic, and dose-volumetric parameters. To determine the key clinical risk factor for RILD, we determined the risk of RILD by fitting a binary logistic regression model based on characteristics such as gender, age, alpha-fetoprotein levels, etiology, Eastern Cooperative Oncology Group performance status, EQD2, and albumin–bilirubin (ALBI) score. To prevent multicollinearity between factors such as ALBI and CP class, we chose only one factor per clinical indicator for analysis. Clinical factors with *p*-values of <0.05 in multivariable analysis were included in the predictive model. Above analyses were performed in SPSS 22 (SPSS, Inc., Chicago, IL, USA).

Considering the curse of dimensionality in radiomic features, we used the K-nearest neighbor (KNN) and the sequential forward selection (SFS) algorithm as the greedy methods to find the best combination of N features associated with RILD. Before the process of feature selection, each feature was transformed into the standardized range (Z-score transformation) based on the mean and standard deviation values of the study cohort.

For dose-volumetric parameters including GTV, normal liver volume, V5, V10, V15, V20, V25, and V30, multivariable analysis is not suitable, as these parameters are interdependent. Accordingly, we also used the KNN and SFS to identify the parameters associated with RILD. Both radiomic and dose-volumetric parameter selections were performed with Python 3.5 (Python Software Foundation, DE, USA).

### 2.6. Model Construction

Random forest (RF) and logistic regression (LR) models were used as classifiers to determine whether the patient may develop RILD after receiving SBRT. The model was computed using Python 3.5. To build this model, we randomly split our dataset (14 RILD cases among 86 patients) at the ratio of 6:2:2 into the training, validation, and test sets. The training set was used to train the model; the extra cases were generated from the minority class in the training set, using the synthetic minority oversampling technique (SMOTE) [18] to modify imbalanced data. We then calculated the possibility score of RILD by fitting the training set to the RF or LR model based on the previously identified predictive factors. The threshold indicative of RILD risk was initially set to 0.5. We subsequently adjusted this threshold and higher-level parameters of the candidate model based on model performance on the training and validation sets to determine the final model. We evaluated the performance of the final model using the test set.

In addition to the model validation, by using the abovementioned hold-out approach, we further applied a 10-fold cross-validation method to validate our model. We randomly split our dataset into 10 subsets with equal sizes. For each run, one of the 10 subsets would be preserved as validation set, while the remaining 9 subsets were used as the training set. In each training set, we applied the SMOTE on the minority class to reduce the potential prediction bias caused by the imbalanced data. This validation process based on the validation set would be repeated for 10 times, and the model performance was calculated by the average and standard deviation across 10 evaluations. The results of comparison between the hold-out and 10-fold cross-validation methods will be provided in the Appendix A.

### 2.7. Statistical Analysis

We evaluated the performance of the predictive model using a confusion matrix with sensitivity, positive predictive value (PPV), specificity, and negative predictive value. In a model that predicts liver toxicity, such as RILD, sensitivity and PPV are the most important indexes; therefore, we used the F1 score to assess model performance. The F1 score was equal to the harmonic mean of sensitivity and PPV. Moreover, both the area under the precision–recall curve (AUPRC) and receiver operating characteristic curve (AUROC) were used to assess output quality. These analyses were all performed with Python 3.5.

## 3. Results

### 3.1. Patient Characteristics

A total of 86 patients were identified, of which 14 developed RILD. There were 14 cases of non-classical RILD, and one of these patients developed both classical and non-classical RILD at the same time. Patient characteristics are summarized in Table 1. Overall, the mean age was 63.2 years, and the median tumor size was 5.6 (range, 1.0 to 20.1) cm. A total of 75 (87.6%) patients had hepatitis B or C, and 32 (37.2%) presented with portal vein thrombosis. Sixty-eight (79%) and 36 (41.9%) patients were CP class A and ALBI Grade 1, respectively. The SBRT dose ranged from 25 to 60 Gy in 2–6 fractions. The median follow-up time was 14.7 (range, 1–105) months for all patients.

### 3.2. Predictive Factors

Multivariable LR analysis revealed that the ALBI score was the only independent predictor of RILD among the clinical factors (Table 2). Besides, we identified a combination of radiomic features associated with RILD using the KNN and SFS algorithms. When the number of the nearest neighbors, K, was equal to three, and the number of features of the best combination, N, was equal to two, we obtained the highest F1 score of 0.762 with the combination of the difference average (GLCM feature) and strength (NGTDM feature). Similarly, among dose-volumetric parameters, using the KNN and SFS method, we identified the combination of V5 and V30 with the highest F1 score (0.678), given K = 2 and N = 2.

### 3.3. Random Forest Model

The procedure of the factor selection identified the ALBI grade, difference average, strength, V5, and V30 as factors associated with RILD. After oversampling the training set using SMOTE, we obtained a dataset of 84 cases with 42 RILD patients. We determined the probability score of RILD in this training set by fitting an RF model to the key predictive factors. The accuracy of the model in the training set was 100%. The AUPRC and AUROC were both 1 (Figure 1A).

Moreover, the accuracy, sensitivity, PPV, and F1 score of the model in the validation set were 88.9, 33.3, 100.0, and 0.5, respectively. The AUPRC and AUROC were 0.739 and 0.800 (Figure 1A), respectively. Based on the relationship between the F1 score and threshold value in the validation set (Figure 1B), we adjusted the threshold from 0.5 to 0.456, which was the best cut-off point; afterward, model accuracy, sensitivity, PPV, and F1 score in the validation set increased to 94.4, 66.7, 100, and 0.8, respectively. Finally, we evaluated the performance of the current model using the test set. The accuracy, sensitivity, PPV, and F1 score were 94.4, 100, 75, and 0.857, respectively (Table 3). The AUPRC and AUROC were 0.764 and 0.956 (Figure 1A), respectively. We also tried to build models without using radiomics, and the results were shown in Figure 1C and Table 3.

### 3.4. Logistic Regression Model

After oversampling the training set using the SMOTE, we fitted an LR-based model with the same key predictive factors (ALBI grade, difference average, strength, V5, and V30). The accuracy of the model in the training set was 89.3% and the F1 score was 0.901. The sensitivity, PPV, AUPRC, and AUROC were 97.6, 83.7, 0.915, and 0.938 (Figure 2A), respectively. The accuracy of the model in the validation set was 94.4% and the F1 score was 0.8. The sensitivity, PPV, AUPRC, and AUROC were 66.7, 100, 0.850, and 0.956 (Figure 2A), respectively. We then determined the relationship between the F1 score and the threshold value (Figure 2B), which revealed 0.531 as the best cut-off point. Finally, we evaluated the final LR model using the test set. The accuracy of the LR model was 94.4%. The sensitivity, PPV, and F1 scores were 66.7, 100, and 0.8 (Table 3). The AUPRC and AUROC were 0.777 and 0.889 (Figure 2A), respectively. We also tried to build the LR model without using radiomics, and the results were shown in Figure 2C and Table 3.

## 4. Discussion

To the best of our knowledge, this is the first study to propose a predictive model of RILD, based on the combination of radiomic, clinical, and dose-volumetric parameters in HCC patients receiving SBRT. The proposed models have a high F1 score, AUPRC, and AUROC for predicting RILD. In the RF-based model, the F1 score, AUPRC, and AUROC were 0.857, 0.764, and 0.956, respectively, while the corresponding values for the LR-based model were 0.800, 0.777, and 0.889, respectively (Table 3).

The prevalence of RILD varies from 10 to 36% in HCC patients [4,5,19], and its poor prognosis is universal. Baseline liver function remains the only established risk factor for RILD; it is captured by CP class [4,6,7] or ALBI grade [20,21]. Other factors including V5 to V40 [8], male sex [5], hepatitis B status [4], and D700 to 900 cc [9] remain controversial. Besides, only a few studies have constructed predictive models based on these risk factors.

Dawson et al. [5] proposed a model that captures the dose-volume relationship of the liver and RILD risk in a 3D conformal RT (3DCRT) with conventional fractionation, using the Lyman–Kutcher–Burman normal tissue complication probability (NTCP) model. The accuracy, F1 score, sensitivity, and PPV values of the model were 87.1, 0.25, 25, and 25%, respectively, in CP class A patients [5,10]. Similarly, Xu et al. generated a modified NTCP model to estimate the risk of RILD in PLC patients treated with hypofractionated (5–6 Gy per fraction) 3DCRT [10]. The accuracy, F1 score, sensitivity, and PPV values of the model were 72, 0.35, 87.5, and 21.9%, respectively, in CP class A patients. Moreover, using a similar patient cohort, Zhu et al. also developed an artificial neural network (ANN) model to predict RILD [22]. However, as a powerful deep learning algorithm, the ANN model requires big datasets. Thus, the performance of this ANN model was not better than that of the modified NTCP model; its accuracy, F1 score, sensitivity, and PPV values were in the range of 79.6 to 88.2, 0.39 to 0.61, 75 to 87.5%, and 26.1 to 41.2%, respectively. The modified NTCP model and the ANN model presented with high sensitivity but low PPV. These findings may not translate into the clinical setting, given the false-positive rate of approximately 70%, which may lead to under-treatment because the risk of RILD is overestimated. Additionally, Su et al. [23] developed models and nomograms to predict radiation-induced hepatic toxicity based on a multivariable logistic regression formula with better sensitivity and specificity. The nomogram was made up of two factors, namely a pre-CP score and a dosimetric parameter, such as V15. However, we knew that the prescribed dose and fraction size may vary in the clinical practice to each HCC case, which makes it difficult to define the identical dose constraints to the normal liver [24], because the impact of “V15” may be different from the perspective of the biologically effective dose. Therefore, we believe that adding radiomics in to the predictive model can increase its stability and accelerate the progress of seeking individualized dose constraints.

Since the traditional NTCP model was not suitable for extreme hypofractionated RT (>5 Gy per fraction), which includes SBRT, our predictive model using the LR or RF algorithm was based on the clinical, dose-volumetric, and radiomic parameters. In addition, to balance the importance of sensitivity and PPV, we evaluated our model using the F1 score. In the RF-based model, the accuracy, F1 score, sensitivity, and PPV were 94.4, 0.857, 100, and 75%, respectively, in the test set. The corresponding test set values in the LR-based model were 94.4, 0.8, 66.7, and 100%, respectively (Table 3). The high sensitivity of the present models suggests this model may detect patients at high risk of RILD before SBRT, allowing us to modify further treatment plans accordingly. Moreover, the high PPV and low false-positive rate of our models may help reduce the risk of under-treatment due to the relatively conservative prescribed dose or small PTV.

The present study differed from previous studies in that we apply radiomic features as predictive factors, which may detect the histopathological changes in the liver through images. The pathogenesis of RILD included complex responses related to vascular changes, increased collagen synthesis, and the sequential activation of growth factors and cytokines, leading to the deposition of the extracellular matrix and liver fibrosis [25,26]. Histopathological changes in classical RILD were similar to those observed in vein occlusive disease [27]. The obstruction of the outer cavity of the hepatic vein was possibly caused by radiation-induced endothelial cell damage [28], and the mechanism of non-classic RILD involved hepatocellular loss, hepatic sinusoidal endothelial death, and the activation of myofibroblast-like hepatic stellate cells [29]. Therefore, radiomic features on simulation CT images may identify the high-risk liver for RILD development, such as the liver with regenerating hepatocytes-loss or initially obstructed veins. This observation could partially account for the higher F1 scores and AUROC when combining radiomic features in our model (Figure 1 and Figure 2 and Table 3). Representative cases in Appendix A also demonstrated that a predictive model with radiomic features helps detect the high-risk patient of RILD more accurately compared to those only judged by the clinical and dose-volumetric characteristics. Besides, we calculated the feature importance score of our RF model by the Shapley additive explanation (SHAP) model [30], which revealed that the influence of radiomics is second only to ALBI (Appendix A). Overall, the ALBI grade was the most important influential feature. Patients with a higher ALBI grade, lower value of difference average, or strength, increased the chances of RILD occurrence. However, V5 showed a nonlinear trend that a low or high feature value may reduce the chance of RILD, but the intermediate values may have a high chance of RILD. Finally, V30 showed less impact on the model decision because of its narrower distribution compared to other features.

This study had several limitations. First, it was a retrospective study and included patients from a single medical center, resulting in a relatively small sample size and the lack of an external validation set. Therefore, to examine our model reliability, we divided our sample into training, validation, and test sets at the beginning. We evaluate the performance of our model by the independent test set. Though the number of HCC patients receiving SBRT is still small compared to those receiving TACE or RFA in clinical practice, the role of SBRT has been more and more important in HCC recently, and the RILD significantly affected the outcome of these patients. Thus, we hoped this study could be at least a pioneer study for RILD prediction with radiomics.

Second, we decided not to apply filters to develop a response map to calculate more radiomic features, because features derived from response maps may lack robustness. However, we may also miss some RILD-related features at the same time. Third, we applied the SMOTE to the training set to redress the class imbalance but oversampling the minority might result in model overfitting. Nevertheless, it seemed that we achieved the acceptable bias-variance trade-off, since the AUROC between different sets was similar (Figure 1 and Figure 2A), which suggested little possibility of overfitting. In addition, most of the performance indexes were comparable between the estimations using cross-validation data and the originally proposed hold-out approach, especially for the random forest model. These results may also indicate that there were no concerns of overfitting in the proposed models (Appendix A). Fourth, the median tumor size was 5.6 cm in our study, which was larger compared to some studies [7,9]. This information may remind readers to be cautious when extrapolating our model to the patient with a much smaller HCC. Thus, to reduce the possible biases, we also included the GTV volume and the dose-volumetric parameters when analyzing the predictive factors of the RILD. It seemed that there was no direct relationship between GTV volume and RILD development in our study.

## 5. Conclusions

In the present study, we identified five key risk factors for SBRT-related RILD: ALBI grade, difference average, strength, V5, and V30. Based on these factors, we developed the first radiomic-based predictive model for SBRT-related RILD in patients with HCC; our RF and LR models had high sensitivity, PPV, and F1 scores. The present model may benefit treatment strategies in clinical practice by detecting patients at high risk of RILD before undergoing SBRT. Our results emphasized the possibility to accurately predict RILD with a radiomic-based model, though it still required a large dataset and prospective validation to confirm the clinical feasibility.

## Figures and Tables

**Figure 1 biomedicines-10-00597-f001:**
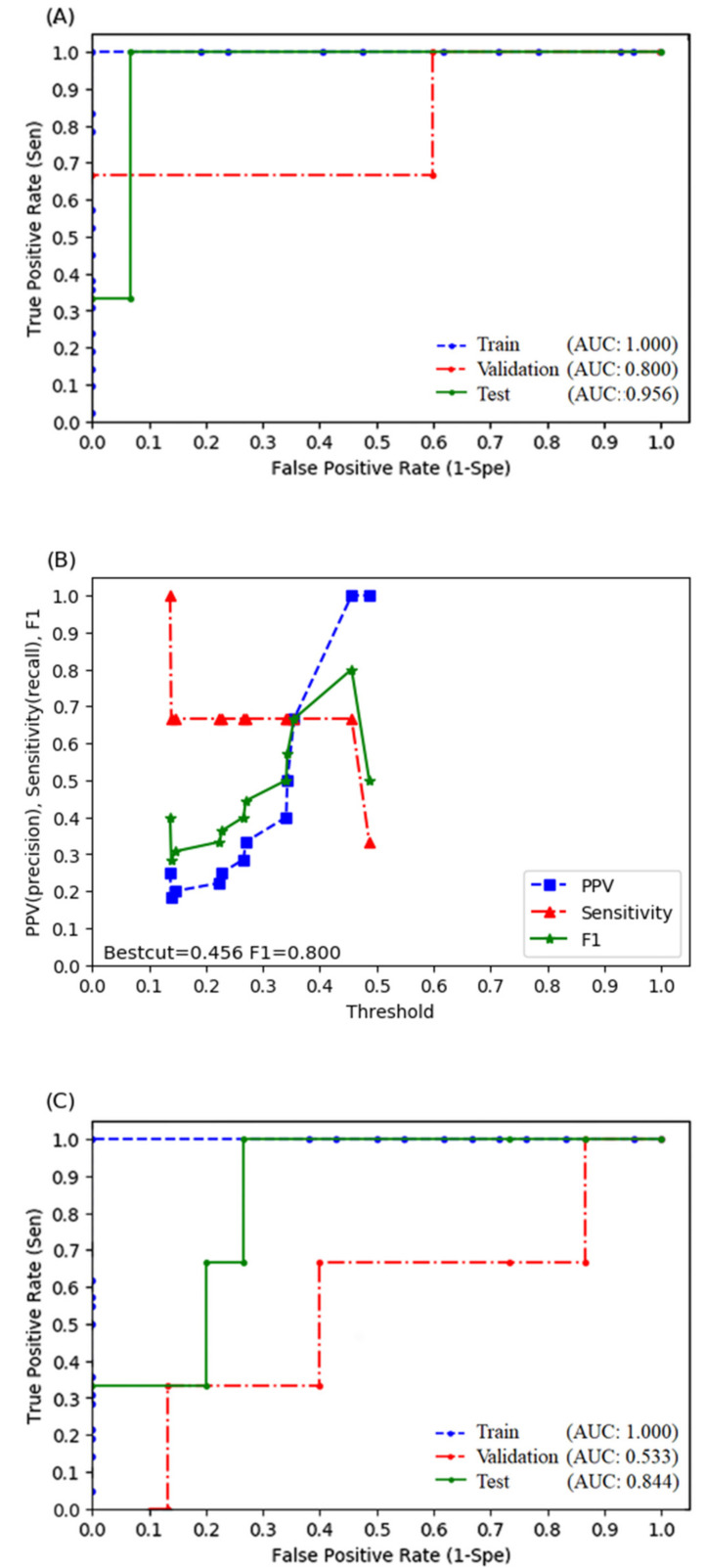
Receiver operating characteristic curve for the random forest (RF) model in training, validation, and test sets with (**A**) or without (**C**) radiomic features. (**B**) The relationship between sensitivity, positive predictive rate, F1 score, and threshold values of the RF model in the validation set.

**Figure 2 biomedicines-10-00597-f002:**
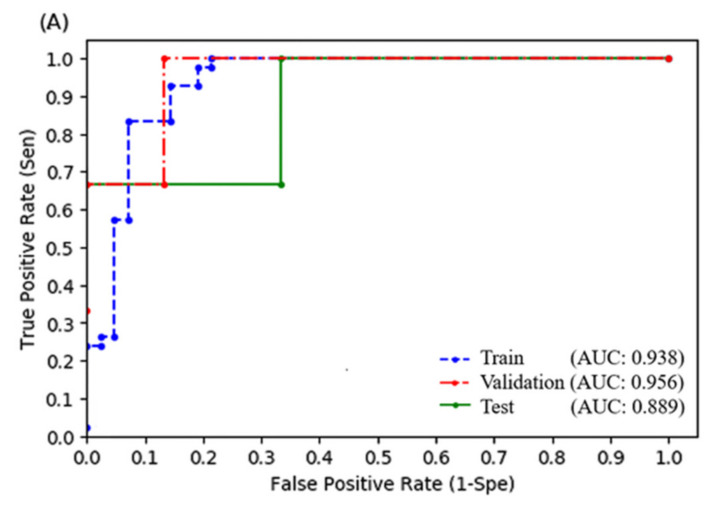
Receiver operating characteristic curve for the logistic regression (LR) model in training, validation, and test sets with (**A**) or without (**C**) radiomic features. (**B**) The relationship between sensitivity, positive predictive rate, F1 score, and threshold values of the LR model in the validation set.

**Table 1 biomedicines-10-00597-t001:** Patient, tumor, and treatment characteristics for overall, RILD and non-RILD cohorts.

Characteristic	All Patient (N = 86)	Without RILD (N = 72)	With RILD (N = 14)
Age (mean)	63.2 ± 12.3	62.9 ± 12.4	65.2 ± 12.2
Sex			
Female	21 (24.4%)	15 (20.8%)	6 (42.9%)
Male	65 (75.6%)	57 (79.2%)	8 (57.1%)
Hepatitis B or C	75 (87.2%)	61 (84.7%)	14 (100%)
With prior tx before SBRT	54 (62.8%)	46 (63.9%)	8 (57.1%)
Tumor size			
Mean, cm	6.4 ± 3.7	6.3 ± 3.9	6.6 ± 2.5
Median (range), cm	5.6 (1–20.1)	5.5 (1–20.1)	6.6 (2–13)
BCLC stage			
0-A	15 (17.4%)	14 (19.4%)	1 (7.1%)
B	11 (12.8%)	9 (12.5%)	2 (14.3%)
C	59 (68.6%)	49 (68.1%)	10 (71.4%)
D	1 (1.2%)	0	1 (7.1%)
ECOG			
0–1	75 (87.2%)	66 (91.7%)	9 (64.3%)
2–4	11 (12.8%)	6 (8.3%)	5 (35.7%)
PVT	32 (37.2%)	24 (33.3%)	8 (57.1%)
CP class			
A	68 (79%)	63 (87.5%)	5 (35.7%)
B	18 (21%)	9 (12.5%)	9 (64.3%)
ALBI grade			
1	36 (41.9%)	35 (48.6%)	1 (7.1%)
2	48 (55.8%)	37 (51.4%)	11 (78.6%)
3	2 (2.3%)	0	2 (14.3%)
Dose	25–60 Gy/2–6 fx	30–60 Gy/2–6 fx	25–55/5 fx
GTV volume			
Mean, cc	217.6 ± 305.0	221.5 ± 325.0	197.9 ± 178.8
Median (range), cc	106.8 (0.7–1817.2)	101.7 (0.7–1817.2)	139.0 (9.4–572.9)
Normal liver volume (mean, cc)	1384.0 ± 522.9	1404.0 ± 515.8	1281.1 ± 566.1
V5 (mean,%)	69.4 ± 20.3	69.0 ± 21.6	71.6 ± 11.9
V10 (mean,%)	44.8 ± 18.6	45.3 ± 19.8	42.3 ± 10.6
V15 (mean,%)	27.1 ± 12.5	27.5 ± 13.2	25.1 ± 8.5
V20 (mean,%)	17.5 ± 8.7	17.7 ± 9.1	16.2 ± 6.5
V25 (mean,%)	11.7 ± 6.5	11.9 ± 6.7	10.6 ± 5.6
V30 (mean,%)	7.7 ± 4.9	7.8 ± 4.9	7.0 ± 4.9
Median f/u (month)	14.7 (1–105)	18.2 (1.7–105)	2.8 (1–58)

RILD: radiation-induced liver disease, tx: treatment, SBRT: stereotactic body radiation therapy, cm: centimeter, BCLC: Barcelona Clinic Liver Cancer, ECOG: Eastern Cooperative Oncology Group, PVT: portal vein thrombosis, CP class: Child–Pugh class, ALBI: albumin–bilirubin, GTV: gross tumor volume, Gy: gray, fx: fractions, cc: cubic centimeter, f/u: follow up.

**Table 2 biomedicines-10-00597-t002:** Analysis of the clinical factor of RILD using logistic regression.

Clinical Variable	Multivariate Analysis		
	HR	95% CI	*p* Value
Gender	0.481	0.010–23.675	0.713
Age	1.041	0.859–1.261	0.683
Pretx AFP	1.000	0.999–1.001	0.576
Etiology			0.952
HBV vs. no	2.256 × 1010	-	0.998
HCV vs. no	4.076 × 1010	-	0.998
HBV and HCV vs. no	1.314 × 1010	-	0.998
ECOG			
2–4 vs. 0–1	57.790	0.613–5444.863	0.080
EQD2	1.018	0.901–1.151	0.770
ALBI score	91.304	2.700–3087.382	0.012

RILD: radiation-induced liver disease, Pretx: pre-treatment, AFP: alpha-fetoprotein, HBV: hepatitis B, HCV: hepatitis C, ECOG: Eastern Cooperative Oncology Group, EQD2: equivalent dose in 2Gy fractions, ALBI: albumin–bilirubin.

**Table 3 biomedicines-10-00597-t003:** Summary of sensitivity, specificity, and accuracy rates for prediction of RILD in the test set by random forest and logistic regression model for hepatocellular carcinoma treated with SBRT.

	Random Forest (With Radiomics)	Random Forest (Without Radiomics)	Logistic Regression (With Radiomics)	Logistic Regression (Without Radiomics)
Sensitivity	1.000	1.000	0.667	0.667
Specificity	0.933	0.600	1.000	0.733
Positive predictive rate	0.750	0.333	1.000	0.333
Accuracy	0.944	0.667	0.944	0.722
F1 score	0.857	0.500	0.800	0.444
AUROC	0.956	0.844	0.889	0.733

RILD: radiation-induced liver disease, AUROC: area under receiver operating characteristic.

## Data Availability

All data generated or analyzed during this study are included in this published article and its Appendix A files. To protect patient privacy, the raw images and data collected in this study can be only accessed by contacting the corresponding authors (C.-S.L. and C.-F.L.).

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
