# Peer review of "Radiomics-Based Predictive Model of Radiation-Induced Liver Disease in Hepatocellular Carcinoma Patients Receiving Stereo-Tactic Body Radiotherapy"

_biomedicines, 2022, doi:10.3390/biomedicines10030597_

Round 1

Reviewer 1 Report

1 –  Page 2 lines 49-51

the role of …… dose-volumetric parameters remains controversial  “  

Among the references mentioned (4-10), only n.7 and n.9 strictly deal with SBRT instead of conventional 3D conformal RT. In view of different pathogenetic mechanisms related to large doses per fraction, I advise great caution in extrapolating results from 3D studies to SBRT.

Also, when taking into account only references dealing with SBRT, in ref. n.9 the predictive role of dose/volume parameters stands out quite clearly, so it seems unfair to use the term controversial.

2 – Page 2 lines 89-90:

Treatment was administered in 2–6 fractions, with a total dose of 25–60 Gy”

I suggest to explain criteria for these large variation, since prescription might be biased by some of the factors evaluated in the study (e.g. GTV volume as well as liver function)

3 - Page 2 lines 90-92:

The median, mean, and range of equivalent dose in 2 Gy fractions (EQD2) with α = 10 Gy were 71.2, 70.2, and 36.4–110.0 Gy, respectively.”

Please add some comment about the choice of  α/β = 10 Gy. Although conventionally used for several tumors, within the field of HCC the evidence for any value seems scarce and also different values have been reported (e.g. in Huang Y et al. Clinical parameters for predicting radiation-induced liver disease after intrahepatic reirradiation for hepatocellular carcinoma. Radiat Oncol. 2016 Jul 2;11(1):89)

Also I suggest some comment about the limits of using EQD2 calculation based on LQ model when dealing with large doses per fraction.

4 – Discussion

The mean GTV volume in this study is remarkably higher than in most reports on SBRT of HCC (see for instance:  references n. 7 and n.9 in this manuscript, as well as several reports described in reviews such as Bang A et al. Radiotherapy for HCC: Ready for prime time? JHEP Rep. 2019 May 21;1(2):131-137, or Mathew AS et al. Current Understanding of Ablative Radiation Therapy in Hepatocellular Carcinoma. J Hepatocell Carcinoma. 2021 Jun 14;8:575-586. 

I suggest to include this difference in patient population as a possible study limitation.

Author Response

Response to Reviewer 1 Comments

Point 1 – Page 2 lines 49-51

“the role of …… dose-volumetric parameters remains controversial  “ 

Among the references mentioned (4-10), only n.7 and n.9 strictly deal with SBRT instead of conventional 3D conformal RT. In view of different pathogenetic mechanisms related to large doses per fraction, I advise great caution in extrapolating results from 3D studies to SBRT.

Also, when taking into account only references dealing with SBRT, in ref. n.9 the predictive role of dose/volume parameters stands out quite clearly, so it seems unfair to use the term controversial.

Response 1:

Thank you for your friendly reminder. To avoid the confusion, we have removed citations related to conventional 3D conformal RT and only cited the studies of SBRT to review the potential role of dose-volumetric parameters. Actually, several studies reported that some dose/volume parameters, including mean liver dose, effective liver volume, and doses to 700-900 cc, were associated with liver toxicity [1-3], but other studies revealed that only clinical factors, such as Child Pugh score  7 could predict the occurrence of RILD [4-6]. Accordingly, we thought that there was no strong consensus nor consistent results on the dose/volume parameters for RILD development. However, we agree the term, ‘controversial’, may be inapprociate, and we have revised the relevant descriptions in the revised manuscipt (Page 2 , Introduction).

References:

[1] Velec M, Haddad CR, Craig et al. Predictors of Liver Toxicity Following Stereotactic Body Radiation Therapy for Hepatocellular Carcinoma. Int J Radiat Oncol Biol Phys. 2017 Apr 1;97(5):939-946. doi: 10.1016/j.ijrobp.2017.01.221

[2] Son SH, Choi BO, Ryu MR, et al. Stereotactic body radiotherapy for patients with unresectable primary hepatocellular carcinoma: dose-volumetric parameters predicting the hepatic complication. Int J Radiat Oncol Biol Phys 2010;78:1073-1080.

[3] Liang SX, Huang XB, Zhu XD, et al. Dosimetric predictor identification for radiation-induced liver disease after hypofractionated conformal radiotherapy for primary liver carcinoma patients with Child-Pugh Grade A cirrhosis. Radiother Oncol 2011;98:265-269.

[4] Jun BG, Kim YD, Cheon GJ et al. Clinical signifi-cance of radiation-induced liver disease after stereotactic body radiation therapy for hepatocellular carcinoma. Korean J Intern Med. 2018;33(6):1093-1102. doi: 10.3904/kjim.2016.412

[5] Jung J, Yoon SM, Kim SY, et al. Radiation-induced liver disease after stereotactic body radiotherapy for small hepatocellular carcinoma: clinical and dose-volumetric parameters. Radiat Oncol 2013;8:249.

[6] Cardenes HR, Price TR, Perkins SM, et al. Phase I feasibility trial of stereotactic body radiation therapy for primary hepatocellular carcinoma. Clin Transl Oncol 2010;12:218-225.

Point 2– Page 2 lines 89-90:

Treatment was administered in 2–6 fractions, with a total dose of 25–60 Gy”

I suggest to explain criteria for these large variation, since prescription might be biased by some of the factors evaluated in the study (e.g. GTV volume as well as liver function)

Response 2: Thank you for your kind suggestion. The prescription dose was determined mainly based on normal organ constraints as our previous publication [1]. Most of recruited patients received 40-50 Gy in 4 to 5 fractions; there was only one patient received 2 fractions, and 2 patients reveived 6 fractions. We tended to give the highest dose to the gross tumor under the allowable dose constraints. For patients with a larger tumor size (> 6 cm), Child Pugh score between 7–8 , or multiple tumors (≥ 3), we would give a more conservative dose. We have added these descriptions into the revised manuscipt (Page 2 , Section 2.2 SBRT).

Radiation Therapy Oncology Group (RTOG) 1112 dose prescription approach suggested a range from 27.5 to 50 Gy in 5 fractions, while National Comprehensive Cancer Network (NCCN) Guildline suggested 30 to 50 Gy in 3 to 5 fractions, and fraction number larger than 5 may also be used if clinically indicated. Bang et al. reviewed several prospective trials of SBRT in HCC with doses ranged from 23 to 75 Gy in 3 to 6 fractions [2]. Accordingly, the doses we administered in the recruited patients were comparable with the prescription doses in these articles and guildlines. In addition, we also included the GTV volume, normal liver volume and the dose-volumetric parameters when analyzing the predictive factors of the RILD to avoid the potential biases.

References:

[1] Huang WY, Jen YM, Lee MS, Chang LP, Chen CM, Ko KH, Lin KT, Lin JC, Chao HL, Lin CS, Su YF, Fan CY, Chang YW. Stereotactic body radiation therapy in recurrent hepatocellular carcinoma. Int J Radiat Oncol Biol Phys. 2012 ;84(2):355-61. doi: 10.1016/j.ijrobp.2011.11.058

[2] Bang A, Dawson LA. Radiotherapy for HCC: Ready for prime time? JHEP Rep. 2019 May 21;1(2):131-137. doi: 10.1016/j.jhepr.2019.05.004.

Point 3–Page 2 lines 90-92:

“The median, mean, and range of equivalent dose in 2 Gy fractions (EQD2) with α/β = 10 Gy were 71.2, 70.2, and 36.4–110.0 Gy, respectively.” :

Please add some comment about the choice of α/β = 10 Gy. Although conventionally used for several tumors, within the field of HCC the evidence for any value seems scarce and also different values have been reported (e.g. in Huang Y et al. Clinical parameters for predicting radiation-induced liver disease after intrahepatic reirradiation for hepatocellular carcinoma. Radiat Oncol. 2016 Jul 2;11(1):89)

Also I suggest some comment about the limits of using EQD2 calculation based on LQ model when dealing with large doses per fraction.

Response 3: Thank you for your question. To our best knowledge, liver tumors may conventionally have a high α/β ratio, though limited data being available in literatures. Some studies, including the report from Huang et al., may use the α/β ratio about 14.7 ± 2 Gy based on the result of Tai et al [1]. This α/β parameter was calculated by fitting their model to the data from Wu et al [2], which composed of a large tumor size (with a median of 10.7 cm). As the result, the authors mentioned that the results obtained from their study should not be used for a patient group with a very different median tumor size [1]. Considering the median tumor size of our patients was around 5.6 cm, we used α/β = 10 Gy to estimate EQD2 as other study did [3].

We appreciate your friendly reminder that the LQ model may over-estimate the biological effect or EQD2 when dealing with a large dose per fraction (>5 Gy). We have added some comments about this issue into the revised manuscipt (Page 3 , Section 2.2 SBRT). We also mentioned simillar viewpoints in the discussion about NTCP model. Finally, we’d like to emphasize that the EQD2 was not used to build the prediction model of RILD in this study.

References:

[1] Tai A, Erickson B, Khater KA, Li XA. Estimate of radiobiologic parameters from clinical data for biologically based treatment planning for liver irradiation. Int J Radiat Oncol Biol Phys. 2008 Mar 1;70(3):900-7. doi: 10.1016/j.ijrobp.2007.10.037.

[2] Wu DH, Liu L, Chen LH. Therapeutic effects and prognostic factors in three-dimensional conformal radiotherapy combined with transcatheter arterial chemoembolization for hepatocellular carcinoma. World J Gastroenterol. 2004 Aug 1;10(15):2184-9. doi: 10.3748/wjg.v10.i15.2184.

[3] Cheung MLM, Kan MWK, Yeung VTY, et al. Analysis of Hepatocellular Carcinoma Stereotactic Body Radiation Therapy Dose Prescription Method Using Uncomplicated Tumor Control Probability Model. Adv Radiat Oncol. 2021 Jun 12;6(5):100739. doi: 10.1016/j.adro.2021.100739.

Point 4–Discussion

The mean GTV volume in this study is remarkably higher than in most reports on SBRT of HCC (see for instance: references n. 7 and n.9 in this manuscript, as well as several reports described in reviews such as Bang A et al. Radiotherapy for HCC: Ready for prime time? JHEP Rep. 2019 May 21;1(2):131-137, or Mathew AS et al. Current Understanding of Ablative Radiation Therapy in Hepatocellular Carcinoma. J Hepatocell Carcinoma. 2021 Jun 14;8:575-586.

I suggest to include this difference in patient population as a possible study limitation.

Response 4: Thank you for your kind suggestion. Although a minority, the studies from Bang et al. and Mathew et al. did include patients with tumors larger than 5 cm, and Mathew et al. also mentioned the role of SBRT in larger HCC. Actually, in the past ten to fifteen years, the role of SBRT in HCC was more like the salvage treatment, which was used when other local treatments were inaccessible or for refractory tumors [1]. Besides, until NCCN guildline version 2017, the external beam RT including SBRT was still a catgory 2B treatment for medically inoperable or unresectable HCC. The present study retrospectively included patients from July 2007 to June 2015, and all of these cases were discussed by the multidisciplinary team. The most common treatment indications of SBRT were the HCC that was medically inoperable or unresectable and unsuitable for TACE or RFA. Patients were informed of the advantages and disadvantages of SBRT, and the final treatment depended on the patients’ decisions. Because small tumors could usually be dealed with radiofrequency ablation or transarterial chemoembolization, a larger median tumor size of 5.6 cm was presented in our study. Our previous study also demonstrated the efficacy of SBRT in these cases [2].

We have added the relevant descriptions to the limitation section into the revised manuscript (Page 10, Discussion). This information may remind readers to be cautious when extrapolating our model to the patient with a much smaller HCC. Besides, to reduce the possible biases, we also included the GTV volume, and the dose-volumetric parameters when analysing the predictive factors of the RILD. It seemed that there was no direct relationship between GTV volume and RILD development in our study.

References:

[1] Park HC, Yu JI, Cheng JC, et al. Consensus for Radiotherapy in Hepatocellular Carcinoma from The 5th Asia-Pacific Primary Liver Cancer Expert Meeting (APPLE 2014): Current Practice and Future Clinical Trials. Liver Cancer. 2016 Jul;5(3):162-74. doi: 10.1159/000367766.

[2] Shen PC, Chang WC, Lo CH, et al. Comparison of Stereotactic Body Radiation Therapy and Transarterial Chemoembolization for Unresectable Medium-Sized Hepatocellular Carcinoma. Int J Radiat Oncol Biol Phys. 2019 Oct 1;105(2):307-318.doi: 10.1016/j.ijrobp.2019.05.066.

Reviewer 2 Report

Introduction

  1. lack of introduction to the SBRT features in cancer prediction
  2. what are those limitations of the current method (line 61)

Method  

  1. Considering the small sample size, instead of splitting the data into three parts, I would recommend validating the performance of the model using cross-validations
  2. when using random forest, it is better to provide the feature importance score, like the shape value

Reviewer 3 Report

The authors present a radiomics-based model, which also includes clinical and dose-volume-histogram parameters, for the prediction of non-classically defined radiation induced liver disease (RILD) in patients treated with stereotactic body radiotherapy (SBRT).

They have performed a multivariate analysis for the determination of the clinical parameters and a radiomics features and dose-volume-histogram parameters selection with K-Nearest-Neighbor (KNN) and Sequential Forward Selection (SFS).

My comments and questions on the paper are the following:

Section 2.3 Patient follow-up

Line 95-96: the authors indicate the acquisition of abdominal dynamic CT and/or MRI scans with a 3-months interval follow-up.

The authors should mentioned how the different images modality acquired at different longitudinal time-point follow-up are analyzed at the radiomics level ?

How the different regions of interest on the CT or MRI were handled? Were the ROI on the MRI scans reported on the simulation CT? How the registration between the MRI and CT was performed?

Section 2.4 Extraction of radiomics features

It is not clear if the radiomics feature extractions is performed only for the ROI on the simulation CT. Even though, for stereotactic body radiotherapy (SBRT) a 4D-CT is required for a respiratory triggered treatment. Please specify how the region of interest used for the radiomics feature extraction from the ROI is managed between the different respiratory phases?

Indeed, my concern is about the dose-volumetric parameters, if the V5 and V30 are retrieved from the calculated SBRT plan, the ROI should also be taken in consideration with the treating respiratory phases in order to get the right percentage of spatial volume of the liver that received the respective dose of 5 and 30 Gy.

Section 3.1. Patient characteristics

Line 177 and Table 1

The median tumor size is given in cm, is it the largest axis of the tumor on one 2D plan? Shouldn’t it be at least a product of the two largest axis given in cm2?

The median tumor should be the volume given by the GTV.

Section 3.3 Random forest model

Line 199: The authors have performed an oversampling of the training set in order to counter balance the minority class and get a dataset of 82 cases with 42 RILD, which represents more than 51%.

Line 236: According to the prevalence that the authors mentioned, RILD varies from 10-36%. The oversampling is too important, isn’t it?

Discussion paragraph

The authors should comment more specifically why V5 (“low dose” percentage) induced more RILD whereas it is high gradient dose that are delivered in SBRT?

Indeed, according to Table 1, the percentage volume (mean ? median? ) of liver which received 5 Gy is more important in the patients with RILD and the percentage volume of liver which received 30 Gy (higher dose) is more important in patients without RILD.

Round 2

Reviewer 2 Report

The author has fully addressed all of my concerns and therefore can be considered for publication.

Reviewer 3 Report

The authors provided supplementary data and gave additionnal explanations that answered my questions and comments.

The manuscript is this revised version can be accepted for publication.